# Tear Break-Up Time and Seasonal Variation in Intraocular Pressure in a Japanese Population

**DOI:** 10.3390/diagnostics10020124

**Published:** 2020-02-24

**Authors:** Masahiko Ayaki, Kazuno Negishi, Kenya Yuki, Motoko Kawashima, Miki Uchino, Kazuo Tsubota

**Affiliations:** 1Otake Clinic Moon View Eye Center, Yamato 2420001, Japan; 2Department of Ophthalmology, Keio University School of Medicine, Tokyo 1608582, Japan; yukikenya@gmail.com (K.Y.); motoko326@gmail.com (M.K.); uchinomiki@yahoo.co.jp (M.U.); tsubota@z3.keio.jp (K.T.)

**Keywords:** dry eye, tear break-up time, intraocular pressure, seasonality

## Abstract

Purpose: To evaluate seasonal variation in intraocular pressure (IOP) with and without short tear break-up time (SBUT, BUT ≤5 s) since dry eye and IOP are known to have seasonal variation. Methods: This study enrolled 176 patients who visited one of six eye clinics, in Japan, four times for IOP measurement, in every season. The mean patient age was 67.9 years, including 79 males. Participants were divided into four groups based on the presence of glaucoma and/or SBUT and we compared the seasonal variation in IOP (winter and summer) among the four groups. Results: The IOP (mmHg) in winter and summer, respectively, was 12.8 ± 3.7 and 12.8 ± 3.1 for non-glaucoma patients without SBUT (*n* = 47, *p* = 0.964), 14.8 ± 3.4 and 13.3 ± 3.4 for non-glaucoma patients with SBUT (*n* = 57, *p* < 0.001), 14.3 ± 3.2 and 14.1 ± 3.4 for glaucoma patients without SBUT (*n* = 36, *p* = 0.489), and 13.3 ± 3.0 and 11.6 ± 2.9 for glaucoma with SBUT (*n* = 36, *p* < 0.001). Seasonal variation was largest across the seasons in the glaucoma with the SBUT group, and the magnitude of seasonal variation correlated with BUT (β = 0.228, *p* = 0.003). Conclusions: Seasonal variation tended to be larger in patients with SBUT than those without SBUT.

## 1. Introduction

Glaucoma and dry eye disease (DED) are common geriatric diseases associated with poor quality of life and requiring ongoing daily topical medication [1,2]. The prevalence of glaucoma and DED in middle aged and elderly Japanese is 4% and 12% to 23%, respectively [3,4], making both disorders a common presentation in daily ophthalmic practice. In addition, glaucoma and DED frequently coexist in a single patient, and some of the glaucoma medications can affect the ocular surface as a side effect [5,6]. For this reason, some glaucoma medications come in preservative-free, single-dose units, with reduced toxicity from chlorhexidine and SofziaR, and a decreased concentration of benzalkonium chloride (BAK), the most commonly used preservative in such agents [7]. DED are also multifactorial diseases that predominantly affect women and the elderly, rheumatoid disease, and immunological disease, and the seasonal variation in symptoms and signs of DED has been well documented [8,9,10,11]. Previous investigations have indicated that different seasons (spring, summer, and winter), cold and hot weather, seasonal air pollen, wind, and sunshine can exacerbate DED. DED patients can have a wide variety of symptoms and ocular surface signs, however, short tear break-up time (SBUT) has been introduced as a representative sign in most recent diagnostic criteria [2].

There are many risk factors for glaucoma in terms of intraocular pressure (IOP) elevation and optic nerve damage [12,13,14,15]. The IOP can also be elevated in subjects with hypertension and in cold weather [16,17,18,19,20,21,22]. Increased melatonin secretion with longer daytimes in summer induces an increase in progesterone androgen secretion resulting in the reduction of IOP [21,23,24,25]. Glaucoma medication is known to suppress the magnitude of seasonal fluctuation of IOP [26,27,28]. 

Clinically, it is unknown whether DED can alter IOP, and therefore glaucoma, since both conditions worsen in winter and both IOP and lacrimal secretion are under neural control [29]. Consequently, IOP control should be given special attention across all seasons. We previously indicated that winter IOP was highest for all seasons in the study of DED signs and symptoms in 1916 cases including glaucoma cases of 5.7% to 13.9% [8]. In the present study, we attempted to evaluate the seasonality of IOP in four study groups stratified for the presence of glaucoma or SBUT, and we compared the seasonal variation in IOP (winter and summer) among the study groups. The obtained results should provide useful information for IOP control in these common diseases. Although we use SBUT for a stratified group name in the present study, the term “short breakup time dry eye” is generally recognized as a specific type of DED characterized with instantaneous instability of the tear film immediately at eye opening.

## 2. Methods

### 2.1. Study Design, Ethical Approval, and Study Population

This study was a multisite, hospital-based, and case/controls with and without glaucoma conducted from March 2015 to February 2017. The data was obtained with a retrospective chart review. The cases and control subjects were recruited from the following six clinical sites: Komoro Kosei General Hospital (Nagano, Japan), Shinseikai Toyama Hospital (Toyama, Japan), Tsukuba Central Hospital (Ibaraki, Japan), Jiyugaoka Ekimae Eye Clinic (Tokyo, Japan), Todoroki Eye Clinic (Tokyo, Japan), and Takahashi-Hisashi Eye Clinic (Akita, Japan).

This study was done on mainland Japan, where the latitude is 35.68 degrees North in the Tokyo area and day length varies by 4 to 6 h over the year according to averages from 1981 to 2010 reported by the Japan Meteorological Agency. Japan has four distinct seasons during which the temperature, humidity, and daylight time markedly vary. Between 1981 and 2010, the average temperature and humidity in the Tokyo area ranged from 5.2 °C and 52% in winter to 25.0 °C and 77% in summer, in Akita from 0.1 °C and 73% in winter to 22.9 °C and 79% in summer, in Toyama from 2.7 °C and 79% in winter to 24.6 °C and 78% in summer, and in Komoro from −0.6 °C and 78% in winter to 23.8 °C and 74% in summer, respectively, as reported by the Japan Meteorological Agency. 

The respective institutional review boards and ethics committees of the Shinseikai Toyama Hospital (permit number 150503) and the Komoro Kosei General Hospital (permit number 2705) approved this study, which was conducted in accordance with the tenets of the 1995 Declaration of Helsinki (as revised in Edinburgh, 2000). Informed consent was obtained from all participants.

Participants were divided into four groups based on the presence of glaucoma or short BUT (≤5 s) as follows; non-glaucoma with normal BUT as a control group, non-glaucoma with short BUT group, glaucoma without short BUT group, and glaucoma with short BUT group.

### 2.2. Inclusion and Exclusion Criteria

Patients were enrolled if they had their IOP measured during four consecutive seasons and received the same medication during the study period. Study participants visited our clinic for regular checkups for early cataract, eye fatigue, DED, and glaucoma. Patients were excluded from the study if they had visual impairment (<20/25 in either eye) or were under 20 years of age. Patients were excluded from the study if they had history of past glaucoma surgery, or any ocular surgery within twelve months. We excluded patients from glaucoma cases if they have secondary glaucoma, uveitis, or steroid administration.

### 2.3. Ophthalmological Examinations

IOP was measured four times (spring, summer, fall, and winter) and the magnitude of seasonality was calculated by subtracting summer IOP from winter IOP in an individual. All patients had IOP measurements for every 2 to 3 months throughout the year and the interval of analyzed IOP for each season was usually 2 to 4 months. IOP was measured in the morning session between 8:00 a.m. and 12:00 midday, using a non-contact tonometer (Tonoref^TM^ II, Nidek Co. Ltd., Aichi, Japan). The mean BUT and corneal staining scores (0–9 points) were based on the Japanese dry eye diagnostic criteria [30], with BUT evaluated three times, and the mean value determined. Corneal fluorescein staining scores were evaluated in three areas and scored on a 0- to 2-point scale (0, no damage to 2, damaged entirely); the scores were summed up to a maximum of 2 points in total. Ocular surface abnormality was defined as a BUT ≤ 5 s and corneal staining scores ≥1 point based on previous studies [31,32]. The BUT and corneal fluorescein staining score were determined at the date of each patient’s visit in 2017 and classified as follows: spring from March to May, summer from June to August, fall from September to November, and winter from December to February. All examinations were performed at a temperature of 18 °C to 25 °C and humidity of 40% to 60%.

All glaucoma cases were bilateral open angle glaucoma or normal tension glaucoma. For glaucoma diagnosis, we conducted a visual field test (Humphrey Visual Field Analyzer 30-2 standard program; Carl Zeiss, Jena, Germany), measuring the thickness of ganglion cell complexes using optical coherent tomography (OCT; RC3000R (Nidek, Gamagori, Japan) and Cirrus^®^ HD-OCT (Carl Zeiss, Jena, Germany)), and then routine examinations were performed. As described previously [23], diagnostic criteria for glaucoma in the present study comprised glaucomatous visual field loss tested using the Glaucoma Hemifield Test, an ophthalmoscopic neurofiber layer defect, a cup/disc ratio >0.6, or elevated IOP (>21 mmHg) requiring topical medication for more than 6 months. We confirmed no change in glaucoma medication during the study period with a chart review. Exclusion criteria were coexisting cataract with significant lens opacity disturbing the optical axis that accounted for subjective visual disturbance or decreased visual function, retinal pathology, retinal surgery, or photocoagulation affecting the visual field. Topical glaucoma medications are listed in Appendix A. 

### 2.4. Statistical Analysis

The effects of each season on the IOP were compared as follows: On the basis of a preliminary study and previous investigations that suggested IOP was highest in winter and lowest in summer, we identified the mean IOP for each season, and then compared summer and winter. We compared the IOP of summer and winter with the individual IOP using paired t-test. We used the Mann–Whitney U test with Bonferroni correction to compare the magnitude of seasonality (winter IOP and summer IOP) among four groups. Regression analysis was performed to explore which parameters mostly affected the magnitude of seasonal difference in IOP including age, gender, glaucoma medication type, medication frequency, medication number, refraction, glaucoma severity, and corneal parameters, since it is unknown which factor can affect seasonal fluctuation of IOP in DED cases. Ophthalmological parameters and glaucoma medications were analyzed with multiple regression from the first analysis since most cases were prescribed with PG (Prostaglandin) as a baseline and beta blocker and carbonic anhydrase inhibitor (CAI) were often concomitantly prescribed. Data are presented as the mean ± standard deviation (SD) or as percentages where appropriate. All analyses were performed using StatFlex (Atech, Osaka, Japan), with *p* < 0.05 considered significant.

## 3. Results

The characteristics, ophthalmological results, and DED-related medications of 176 patients enrolled in this study (79 males and a mean age of 67.9 years) are detailed in Table 1. The corneal staining score of short BUT groups (non-glaucoma with short BUT and glaucoma with short BUT) was higher than that of the control (*p* = 0.018 and *p* = 0.002, respectively). Non-glaucoma with short BUT group was prescribed more dry eye medications as compared with the other groups; more than half (56%) of them received DED-related medications and 17% to 31% of the patients in the other groups. Steroid eye drops were used only for non-glaucoma groups. The distribution of seasons for BUT determination was similar in short and normal BUT groups (*p* = 0.305 for non-glaucoma groups and *p* = 0.118 for glaucoma groups, Mann–Whitney test) and there was a difference between the glaucoma and non-glaucoma groups (*p* = 0.010, Kruskal–Wallis test). 

Prostaglandin analogues provided a first-line medication for 97.2% (70/72) of glaucoma patients and beta-blockers were prescribed for 25 cases (34.7%, Appendix A). The mean number of eye drops was 1.4 ± 0.7 for glaucoma without short BUT group and 1.2 ± 0.5 for glaucoma with short BUT group (*p* = 0.342, Kruskal–Wallis test), and the mean frequency of instillation was 1.4 ± 1.0 for glaucoma without short BUT group and 1.7 ± 1.3 for glaucoma with short BUT group (*p* = 0.416). Monotherapy was administered for 28 cases (77.8%) of glaucoma without short BUT group and 27 cases (75.0%) of glaucoma with short BUT group. All of the prescribed glaucoma medications contained benzalkonium chloride or another preservative; SofZiaR in TravatanzR and PolyquadR in DuotravR. 

The mean IOP was significantly different between winter and summer in the short BUT groups (*p* < 0.001 for both glaucoma and non-glaucoma groups, paired t-test), but not for the normal BUT groups (*p* = 0.964 for control group and *p* = 0.489 for glaucoma group) (Table 2). The mean magnitude of seasonality (winter and summer) was greater in the short BUT groups than the control, and it was largest in the glaucoma with short BUT group. A comparison of IOP across seasons between glaucoma groups revealed significantly lower IOP in the short BUT group for all seasons except winter.

The individual range of IOP was smaller for the glaucoma groups than the non-glaucoma groups as indicated in minimum and maximum interindividual variation across four seasons; 10 and 10 mmHg for glaucoma groups with and without short BUT, respectively, and 17 and 20 mmHg for non-glaucoma groups. The graphic representation of the magnitude of seasonality are shown in Figure 1 and Appendix A.

The mean ± SD value of IOP in each group presenting for IOP measurements was higher in winter than in summer. The IOP values at the same letter were compared and study groups with short tear break-up time (sB) showed larger fluctuation than those groups without sB. *p** < 0.05, paired t-test; *P*^a^ = 0.964; *P*^b^ < 0.001; *P*^c^ = 0.489; and *P*^d^ < 0.001.

The linear regression analysis revealed that the magnitude of seasonality in IOP was correlated with BUT and the number and frequency of medication, whereas it was not correlated with age, sex, corneal staining score, mean deviation, refractive errors, and the type of medication (Table 3). The multiple regression analysis demonstrated that BUT was most strongly correlated with seasonality among three variables.

## 4. Discussion

The present results indicated the following two major findings: Seasonality of IOP was greater with a short BUT as compared with normal BUT, and IOP was lower with short BUT as compared with normal BUT. According to the reported seasonal variation of DED [8], the Schirmer test value was the worst in the winter, and BUT and corneal staining scores were the second worst in the winter for DED patients. In contrast, all of the corneal signs were less severe in summer. It is notable that seasonal variation in IOP in our study was similar to that reported for corneal staining score in a DED group described previously [8]. It could be hypothesized that corneal damage induces inflammation on the ocular surface leading to IOP increases mediated by bioactive molecules including TGF beta [33,34] and prostaglandins [35]. Distress associated with uncomfortable symptoms including irritation, pain, and dryness could also raise IOP, particularly because depression is prevalent in DED patients and worsens in winter, and thus this state could result in ocular hypertension [36,37,38,39]. Such an outcome could also explain the IOP reduction noted in summer when the corneal staining scores and DED symptoms are least severe. IOP and blood pressure have been positively correlated and increased in winter [14,15], and adrenergic receptor activation has been conventionally proposed as a possible regulatory system for this phenomenon [40]. Most glaucoma medications contain preservative(s) for infection control and there was no difference in preservatives between the glaucoma groups with and without SBUT (Appendix A). Their ocular surface toxicity and bactericidal effects did not change across seasons and were not associated with IOP fluctuation. IOP fluctuations are generally differentiated in short-term (diurnal) and long-term IOP fluctuations (months vs. years) and the seasonal IOP fluctuation is the latter one. Our results suggested it could be present in both glaucoma and non-glaucoma cases. 

The present results indicated that IOP was lower in cases with short BUT than in those with normal BUT, and this difference was larger between glaucoma groups than between non-glaucoma groups. Taken together, lower IOP values and larger winter IOP rises were predominantly observed in glaucoma cases with short BUT. This is paradoxical since if IOP increases with worsening of DED in the cold and dry weather of winter, IOP in short BUT should be higher than in normal BUT for all seasons. A possible explanation for this seeming anomaly is cornel thinning and drug penetration facilitated by disruption of the ocular surface barrier effects in DED, as previous investigations have suggested that IOP estimated as a lower value in thin cornea and intraocular drug effects of instilled eye drops depends on the drug penetration [12,41,42]. Sleep disorder, depression, and possible decreases in melatonin secretion could also contribute to the dysregulation of IOP [21,23,24,25] in winter when sunshine decreases and DED patients suffer more stress and worsened symptoms. 

Our study has several limitations. First, because this is an observational study, unmeasured or residual confounding factors could remain. For example, additional studies performed with measurement of aqueous tear production are warranted to investigate the association between the aqueous tear deficiency type DED and IOP fluctuation, although this study focused on evaporative DED by using BUT as a major parameter of DED in accordance with the latest diagnostic criteria [2]. Nonetheless, our multivariable adjusted analyses of major possible contributory factors should attenuate potential errors. Second, there is a fundamental limitation in the lack of data for corneal thickness and blood pressure because we failed to confirm whether thinner corneal thickness in DED including cases with short BUT and seasonal variation in blood pressure were directly attributed to the seasonal variation in IOP or other relevant neuronal factors such as use of neuronal or psychiatric medications in glaucoma cases. Adrenergic receptor activation is the most widely accepted regulatory system for IOP seasonality, therefore, future longitudinal studies with systemic evaluations of neuronal aspects are necessary to gain an understanding of seasonal IOP variation in glaucoma patients with short BUT. Third, because the majority of our participants were Japanese with normal tension glaucoma, our data lacked generalizability. Hence, additional studies performed in patients with different ethnicities and glaucoma types are warranted to investigate the association between seasonality and BUT. Thus, we interpreted age- and sex-adjusted and multivariable-adjusted models in addition to univariable model results with some caution. Fourth, we could have used Goldmann tonometer as a gold standard for IOP measurement; however, we considered the IOP data obtained for our study were accurate because most examinations were performed by certified orthoptists (national licensure) and we analyzed the mean IOP of three measurements. Finally, potential selection bias and heterogeneity among intergroups could not be completely eliminated, although the distribution of baseline characteristics for age and sex were reasonable in that more women were included in the short BUT groups and more myopic subjects were included in the glaucoma groups. In addition, our study examined BUT once at the time of study entry and BUT values can vary among seasons on an individual basis. Thus, BUT and IOP could be measured in each season and a mixed effect model applied, although the distribution of the season for BUT measurement was uniformly distributed in the present study enabling us to address this issue. 

This study has several strengths. First, this study was conducted in Japan with four distinct seasons presenting various temperatures and humidity fluctuations. Second, the samples were collected from multiple institutions in Japan, allowing us to conduct a case-control study including enriched ophthalmic parameters in a rigorous manner. The novelty of our current study is that we successfully captured the distinctly enhanced seasonal variation in glaucoma patients accompanied with short BUT-type DED, being very common and severely affecting the quality of life throughout eye disease. Additionally, the present results were demonstrated as comparable with those of a large-scale study with over 4000 cases to minimize a variety of bias [43]. In a large-scale study we examined each patient once to immediately correlate IOP with DED, and the present study was carried out in a standard method with repeated examinations on the same patients to adequately analyze seasonal variation.

In conclusion, the eye care specialists should be careful of IOP in glaucoma cases with short BUT since seasonal IOP fluctuation can be greater than normal BUT cases and special attention should be paid for IOP rise in winter. BUT measurement could be recommended in glaucoma cases since any glaucoma medication can cause or aggravate DED and it could be associated with considerable IOP fluctuation.

## Figures and Tables

**Figure 1 diagnostics-10-00124-f001:**
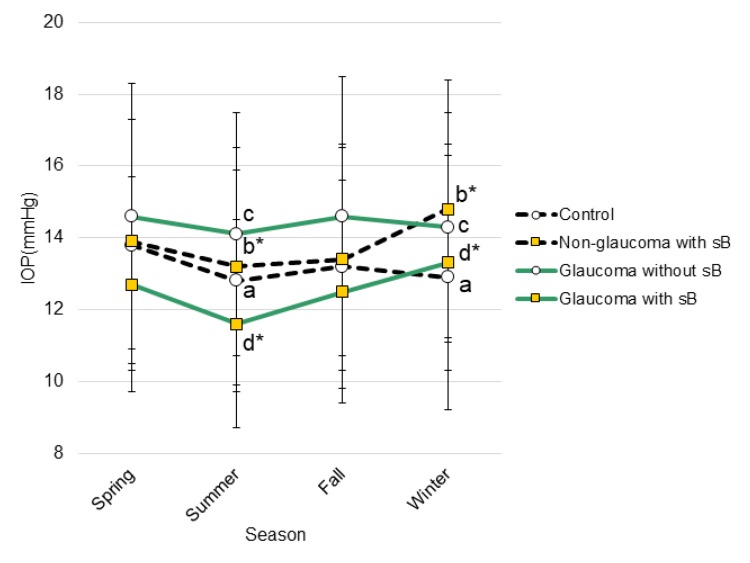
Seasonal variation in intraocular pressure (IOP).

**Table 1 diagnostics-10-00124-t001:** Demographics and dry eye medication in each group.

	Control	Non-Glaucoma with sB	Glaucoma without sB	Glaucoma with sB
# Participants	47	57	36	36
Age (years)	71.5 ± 14.1	66.5 ± 15.3 (0.144)	64.1 ± 13.8 (0.028)	71.1 ± 13.6 (1.000)
% of men	44.7	24.6 (0.030)	75.0 (0.010)	47.2 (0.818)
Refractive errors (diopter)	-0.68 ± 2.78	-1.41 ± 2.85 (1.000)	-2.77 ± 3.64 (0.022)	-1.96 ± 3.40 (0.191)
# eyes with IOL (0/1/2)	37/6/4	44/3/10	30/2/4	24/4/8
Corneal staining score	0.09 ± 0.35	0.37 ± 0.61 (0.018)	0.17 ± 0.38 (0.913)	0.53 ± 0.74 (0.002)
Dry eye medication (%)	**Control**	**Non-Glaucoma with sB**	**Glaucoma without sB**	**Glaucoma with sB**
None	76.6	43.9 (0.001)	83.3 (0.451)	69.4 (0.464)
Hyaluronate	10.6	21.1	11.1	19.4
Mucin secretagogue	14.9	35.1	5.6	11.1
Steroid	4.3	10.5	0	0
Season of visit to determine BUT				
Spring	0	1	3	5
Summer	44	48	16	22
Fall	2	1	6	4
Winter	1	7	11	7
Glaucoma parameters				
Mean deviation (dB)			−4.46 ± 5.76	−5.61 ± 5.14 (0.386)
Disc cupping (%)			73.2 ± 16.8	75.7 ± 17.8 (0.957)

*p* value compared with control in parentheses, using the tests for significance were the Mann–Whitney U test with Bonferroni correction for continuous variables and the Chi-square tests for categorical variables. Abbreviations: sB, short tear break-up time; BUT, tear break-up time; IOL, intraocular lens. # Participants as Number of participants. # Eyes as Number of eyes.

**Table 2 diagnostics-10-00124-t002:** Mean intraocular pressure (mmHg) in each season.

	Control	Non-Glaucoma with sB	Glaucoma without sB	Glaucoma with sB	*p* Value ^C^	*p* Value ^D^
Spring	13.8 ± 3.3	13.9 ± 3.5	14.6 ± 3.3	12.7 ± 3.0	0.973	0.021
Summer	12.8 ± 3.1	13.3 ± 3.4	14.1 ± 3.4	11.6 ± 2.9	0.747	0.001
Fall	13.2 ± 3.1	13.4 ± 3.4	14.6 ± 3.9	12.5 ± 3.1	0.844	0.008
Winter	12.8 ± 3.7	14.8 ± 3.4	14.3 ± 3.2	13.3 ± 3.0	0.218	0.112
*p* value (summer vs winter)	0.964	<0.001	0.489	<0.001		
Mean magnitude of seasonality ^A^ (mmHg)	0.0 ± 3.2	1.5 ± 2.8 (0.023)	0.2 ± 2.1 (1.000)	1.7 ± 2.3 (0.032)	0.091	0.009
Highest seasonality ^B^	8	7	5	6		
Lowest seasonality	−12	−10	−5	−4		
Range (highest-lowest)	20	17	10	10		

^A^*p* value as compared with control in parentheses, using Mann–Whitney U test with Bonferroni correction; ^A^ = (winter and summer); ^B^ = maximum value of the magnitude; ^C^ = control vs non-glaucoma with sB; ^D^ = glaucoma without sB vs. glaucoma with sB. Abbreviations: sB, short tear break-up time.

**Table 3 diagnostics-10-00124-t003:** Regression analysis of the magnitude of IOP and parameters.

	Linear Regression		Adjusted for Age and Sex	
	β	*p*-Value	β	*p*-Value
Age	−0.077	0.519		
Sex ^A^	0.019	0.874		
Corneal parameters				
Tear break-up time (sec)	−0.224	0.003 **	−0.237	0.002 **
Corneal staining score	0.100	0.403	0.126	0.107
Glaucoma-related parameters				
Beta blocker	0.084	0.581	0.067	0.662
Carbonic anhydrase inhibitor	0.128	0.399	0.156	0.320
Number of medications	0.269	0.022 *	0.266	0.025 *
Frequency of medication	0.246	0.037 *	0.247	0.039 *
Refractive error (D)	0.007	0.929	−0.044	0.569
Mean Deviation (dB)	−0.123	0.320	−0.136	0.281
Disc cupping (%)	−0.063	0.617	−0.049	0.712
**Multiple Regression**	**Non-Adjusted**		**Adjusted for Age and Sex**	
Tear break-up time	−0.292	0.011 *	−0.356	0.004 **
Number of medications	0.372	0.346	0.237	0.572
Frequency of medication	−0.141	0.720	−0.013	0.973

^A^ male = 1 and female = 0. * *p* < 0.05 and ** *p* < 0.01. Abbreviations: CAI, carbonic anhydrase inhibitor; NFL, thickness of peripapillary nerve fiber layer.

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
