# Peer review of "Tear Break-Up Time and Seasonal Variation in Intraocular Pressure in a Japanese Population"

_diagnostics, 2020, doi:10.3390/diagnostics10020124_

Round 1

Reviewer 1 Report

The study aims to evaluate seasonal variation in intra-ocular pressure (IOP) with and without short tear break-up time (SBUT, BUT ≤ 5 s). The choice of the pathological conditions is well motivated as glaucoma and dry eye (DED) often happen simultaneously not least because of the adverse effects of glaucoma medications. 176 patients were enrolled from six eye clinics in Japan. Participants were divided into four groups based on the presence of glaucoma and/or short BUT (≤ 5 seconds): non-glaucoma with normal BUT as control; non-glaucoma with short BUT; glaucoma without short BUT; glaucoma with short BUT. Number of ocular surface clinical parameters and possible related and confounding factors (medical treatment history, sex of the patients, etc.) were continuously evaluated. Linear regression analysis revealed that the seasonality in IOP was correlated with BUT and the number and frequency of medication, whereas it was not correlated with age, sex, corneal staining score, mean deviation, refractive errors, and the type of medication. Multiple regression analysis demonstrated BUT was most strongly correlated with seasonality among three variables. The age-/sex- adjusted and multivariable- adjusted models confirm the above described trends. The main conclusion is that the eye care specialists should be aware of IOP in glaucoma cases with short BUT since seasonal IOP fluctuation may be greater than normal BUT cases and special attention should be paid for IOP rise in winter. BUT measurement are to be recommended since any glaucoma medication may implicate DED which in turn could be associated with significant IOP fluctuation.

The limitations and the strengths of the study are thoroughly and objectively discussed in lines 210-250 of the manuscript and well justify the merit of the current study as a pioneer attempt to initiate further detailed research on the topic.

The following relatively minor points should be addressed in the manuscript prior publication:

- In the Introduction, lines 33-34, it is stated that “…. For this reason, glaucoma medication comes in a preservative-free, single dose unit, with  reduced toxicity from chlorhexidine and SofziaR, and a decreased concentration of benzalkonium chloride (BAK), the most commonly used preservative in such agents”. Please support this sentence with references that specifically compare the mentioned preservative components like https://doi.org/10.1039/C3SM51849C, https://doi.org/10.1167/iovs.10-6271, https://doi.org/10.1167/iovs.12-9907  etc.

- In numerous recent high profile studies of tear film dynamics in health and disease the term “short breakup time dry eye” was coined to a specific type of DED characterized with instantaneous instability of the tear film immediately at eye opening. Here SBUT appears to be a general measure for DED. This should be briefly discussed or depending on the authors preferences some other term like “rapid BUT’ or “accelerated BUT” etc. can be implemented to avoid confusion.

- Please in an online supplementary file provide the residuals plots from the regression analyses or some other summary statistics which reports on the conformance to the homoscedascity assumption.  P values alone are not sufficiently informative on this.

- A very minor purely technical point. In the author affiliation sections the line “2 Affiliation 2; [email protected]” should be edited.

Author Response

diagnostics-711569

Thank you very much for reviewing our manuscript. To aid in the re-review of this manuscript, we have included a point-by-point response to each comment. The comments are italicized and placed in square brackets.. In addition, within the revised manuscript, we have used colored and underlined text to highlight changes in response to the reviewers’ comments.

Kazuno Negishi, MD

Masahiko Ayaki MD

  •  
  • Reviewer comments:

    Reviewer #1

    [- In the Introduction, lines 33-34, it is stated that “…. For this reason, glaucoma medication comes in a preservative-free, single dose unit, with  reduced toxicity from chlorhexidine and SofziaR, and a decreased concentration of benzalkonium chloride (BAK), the most commonly used preservative in such agents”. Please support this sentence with references that specifically compare the mentioned preservative components like https://doi.org/10.1039/C3SM51849C, https://doi.org/10.1167/iovs.10-6271, https://doi.org/10.1167/iovs.12-9907  etc.]

We appreciate the comment. We added reference 7 as suggested.

Georgiev, GA, Yokoi N, Ivanova S, Krastev R, Lalchev Z. Surface Chemistry Study of the Interactions of Pharmaceutical Ingredients with Human Meibum Films. Inv Ophthalmol Vis Sci 2012; 53: 4605-4615.

[- In numerous recent high profile studies of tear film dynamics in health and disease the term “short breakup time dry eye” was coined to a specific type of DED characterized with instantaneous instability of the tear film immediately at eye opening. Here SBUT appears to be a general measure for DED. This should be briefly discussed or depending on the authors preferences some other term like “rapid BUT’ or “accelerated BUT” etc. can be implemented to avoid confusion.]

We appreciate the comment. We amended the title and introduction as follows;

Title

Short Tear Break-Up Time and Seasonal Variation in Intra-Ocular Pressure in a Japanese Population”

Introduction, page 2

“Although we use SBUT for a stratified group name in the present study, the term “short breakup time dry eye” is generally recognized as a specific type of DED characterized with instantaneous instability of the tear film immediately at eye opening.”

[- Please in an online supplementary file provide the residuals plots from the regression analyses or some other summary statistics which reports on the conformance to the homoscedascity assumption.  P values alone are not sufficiently informative on this.]

We appreciate the comment. We agree with the reviewer’s suggestion and we wish to have provided the residuals plots from the regression analyses or some other summary statistics which reports on the conformance to the homoscedascity assumption, however, we are sorry that further information is not available.

[- A very minor purely technical point. In the author affiliation sections the line “2 Affiliation 2; [email protected]” should be edited.]

We appreciate the comment. We amended the title page accordingly.

Reviewer 2 Report

The authors assessed seasonal variations in intraocular pressure in healthy subjects and glaucoma patients with and without short tear break up time. Please find my comments below:

1. Page 1, Line 33: The statement regarding the use of single dose units for glaucoma treatment needs to be rewritten, otherwise it appears to the reader that there are only preservative-free treatments available, which is not correct, since several glaucoma patients still receive glaucoma medication containing preservatives.

2. As the authors state correctly, BUT was not measured at every visit but only once at inclusion. This is a major limitation of the manuscript, since it could be possible that there were large fluctuations in BUT. Since these data are not available, a correlation analysis for BUT and IOP at the timepoint both parameters were assessed among all groups would be of interest.

Author Response

diagnostics-711569

Thank you very much for reviewing our manuscript. To aid in the re-review of this manuscript, we have included a point-by-point response to each comment. The comments are italicized and placed in square brackets.. In addition, within the revised manuscript, we have used colored and underlined text to highlight changes in response to the reviewers’ comments.

Kazuno Negishi, MD

Masahiko Ayaki MD

  •  
  • Reviewer comments: 

    Reviewer #2

[1. Page 1, Line 33: The statement regarding the use of single dose units for glaucoma treatment needs to be rewritten, otherwise it appears to the reader that there are only preservative-free treatments available, which is not correct, since several glaucoma patients still receive glaucoma medication containing preservatives.]

We appreciate the comment. We amended the text as follows;

Introduction, page1

“For this reason, some of glaucoma medication comes in a preservative-free,,,,”

[2. As the authors state correctly, BUT was not measured at every visit but only once at inclusion. This is a major limitation of the manuscript, since it could be possible that there were large fluctuations in BUT. Since these data are not available, a correlation analysis for BUT and IOP at the timepoint both parameters were assessed among all groups would be of interest.]

We appreciate the comment. Further investigations should be conducted to explore a

correlation analysis for BUT and IOP at the timepoint both parameters were assessed      among all groups.

Reviewer 3 Report

Scientifically well performed study.  Presentation is clear and well written. The topic is specifically of interest to ophthalmologists and eye researchers and supports that tear film behavior may have seasonality variability. However there appears to be no mention of the humidity and temperature for the examination rooms where the measurements were taken.  Unless this is held reasonably constant through the seasons the measurements could be affected. The authors should address this.  However this should not affect the suitability of the paper for publication.

Author Response

diagnostics-711569

Thank you very much for reviewing our manuscript. To aid in the re-review of this manuscript, we have included a point-by-point response to each comment. The comments are italicized and placed in square brackets.. In addition, within the revised manuscript, we have used colored and underlined text to highlight changes in response to the reviewers’ comments.

Kazuno Negishi, MD

Masahiko Ayaki MD

  •  
  • Reviewer comments:

Reviewer #3

[Scientifically well performed study.  Presentation is clear and well written. The topic is specifically of interest to ophthalmologists and eye researchers and supports that tear film behavior may have seasonality variability. However there appears to be no mention of the humidity and temperature for the examination rooms where the measurements were taken.  Unless this is held reasonably constant through the seasons the measurements could be affected. The authors should address this.  However this should not affect the suitability of the paper for publication.]

We appreciate the comment. We amended the Method as follows;

 Methods, page 3

“All examinations were performed at a temperature of 18ºC-25ºC and humidity of 40%-60%.”

Round 2

Reviewer 2 Report

I have no further comments.

This manuscript is a resubmission of an earlier submission. The following is a list of the peer review reports and author responses from that submission.

Round 1

Reviewer 1 Report

The authors significantly improved their manuscript. No further modifications are required.

Reviewer 2 Report

It's novel, but there's no scientific basis.

Despite being multisite, the total number is too low.

Glaucoma and DED should be classified in detail.

Reviewer 3 Report

Comments

Title: it may be useful to include “in a Japanese population “ to the end of the title when we are talking about seasonal variation.

Introduction:

Line 44 “There are many risk factors for glaucoma in terms of ocular hypertension and optic nerve damage,[11- 14] with seasonality in intra-ocular pressure (IOP) well documented as well as higher IOP with hypertension and cold weather.[15-21] “

This sentence could benefit by re-wording. Optic nerve damage is not a risk factor for glaucoma but in fact is the result of glaucoma. I would separate the seasonality of intraocular pressure into a separate more descriptive sentence or two since it is one of the main tenants of the paper.

Line 53 “…and the results suggested DED cases may have significant seasonal IOP fluctuation that has not been described before. “

This is confusing since line 39 states that the seasonal variation in the symptoms of DED has been well documented.

Line 57 The obtained results should provide useful information for IOP control in these common diseases.

The results may help provide information on IOP behavior ? but probably do not give us information on IOP control.

METHODS

  • It is not clear to me from the description (lines 60-64) whether the study was prospective or retrospective.
  • Patients had their eye pressure checked during four consecutive seasons but how many times per season. Did some patients have more than one measurement per season and if so were all measurements included. If there were multiple measurements and only one per season was included, how was it chosen.
  • Did the exclusion of previous glaucoma surgery include previous laser treatment ?
  • Did any pateints have previous cataract surgery or previous laser refractive surgery which may also have impacted their dry eye.
  • What type of non-contact tonometer was used ?
  • It is stated that all glaucoma cases were bilateral open angle or normal tension but only secondary glaucoma, uveitis or steroid glaucoma were excluded. Were there any cases of angle closure or other types of glaucoma ?

Statistical Analysis

Can the authors comment on whether their study contained an appropriate number of patients to have statistical power to measure these differences. For example, there is only a 2 mmHg or less difference between IOP in the different seasons. Are there enough patients in each group to say that difference in IOP by season is meaningful ?

Results

Can the authors comment on why the average IOP is not higher in the glaucoma groups compared to the non-glaucoma groups. Might this be related to a high prevalence of normal tension glaucoma in this Japanese population ? this may again point to a lack of generalizability of this data to other populations.

Table 3. the presentation of the regression analysis could be more clear. It is listed as model 1 and model 2 which are not listed elsewhere. Might be better to list corneal parameter and then TBUT and Corneal Staining indented. Same with glaucoma parameters and then the different drsop indented. Refractive error was not listed in the text or elsewhere as something that was collected so I was surprised to see it. Finally a few of the parameters got to p<0.01 and should have some other notation like a double **.

Discussion

Several theories for why seasonality may effect IOP variations are presented in succession but I am not sure that the study presented evidence of any of these. It may make the paper more clear and concise to focus on one of these explanations and develop it further, perhaps presenting how a future study could be designed to answer such a question.

OTHER:

Some portions of the manuscript are double spaced and others single spaced- I am sure this would be corrected in the final editing.